# Understanding the roots: Local stakeholders' insights on the causes and challenges in combating child marriage in mountainous Karnali, Nepal

**Nished Rijal**[1,2], **Behnoush Ahranjani**[1,3*], **Padma Hitan**[4], **Sian Fitzgerald**[1]

**1** HealthBridge Foundation of Canada, Ottawa, Ontario, Canada, **2** The University of Ottawa, Ottawa, Ontario, Canada, **3** Cumming School of Medicine, University of Calgary, Calgary, Canada, **4** International Nepal Fellowship, Kalikot, Nepal

* bahranjani@healthbridge.ca

## Abstract

Child marriage remains a pressing issue in Nepal despite a decreasing trend in recent decades. The Nepal constitution prohibits child marriage, and the local governments have implemented various strategies to combat it. Nevertheless, child marriage practice continues, particularly in remote and mountainous regions, posing a challenge to the effectiveness and sustainability of interventions. This study aimed to gather insights from key community stakeholders on the consequences and perceived causes of child marriage, which can serve as a foundation for developing culturally appropriate interventions in Karnali province. Eleven Focus Group Discussions and 13 Key Informant Interviews were conducted with a total of 121 participants, with adult and adolescent participants' mean ages of 35.2 and 15.2 years, respectively. The participants were asked about the significance of child marriage in their community, its consequences, and perceived causes. Discussions were transcribed verbatim in Nepali, and a thematic approach was used for data analysis. There was agreement that child marriage has been declining in the community but continues to be a substantial public health issue. Clear comprehension of the consequences of child marriage, both immediate and long-term, was evident. Child marriage was identified to be influenced by a complex interplay of factors across individual, interpersonal, community, and policy levels. Cultural and gender norms and socioeconomic determinants emerged as interconnected and primary causes. An additional theme highlighted the inequity in accessing information and services, particularly for women and girls. The significance of recognizing the agency of individuals and community groups in their ability to change was also underscored. These findings suggest the stakeholders' awareness of child marriage in their community is significant. To develop and implement effective and sustainable interventions, it is crucial for diverse stakeholder groups to take ownership and actively participate in the planning and execution of interventions led by local government initiatives.

**Data availability statement:** This article contains excerpts from qualitative data gathered through FGDs and KIIs. During the ethical review process, the authors did not specify that the data would be shared in a public repository. Making the full transcriptions publicly available would, therefore, conflict with the ethical approval received, as well as with the information given to participants regarding data confidentiality and usage. Furthermore, considering the community's small size and the distinct roles of participants, there is a risk that individuals could be identifiable to others familiar with the community. As such, the complete transcripts will not be shared publicly. Researchers with specific inquiries regarding the data may contact the corresponding author (bahranjani@healthbridge.ca). Alternatively, researchers may contact Cassandra J. Morris, Chair of the HealthBridge Research Ethics Board, at cmorris@healthbridge.ca.

**Funding:** This work was supported by Global Affairs Canada (Contribution agreement number P007889, dated 16 August 2021, awarded to HealthBridge through Sian Fitzgerald). The funders had no role in study design, data collection and analysis, decision to publish, or preparation of the manuscript. No authors received a salary from the funder.

**Competing interests:** The authors have declared that no competing interests exist.

## Introduction

Child marriage, defined as formal or informal marriage before the age of 18, remains a significant global issue. In 2023, the United Nations Children's Fund reported that 640 million girls and women worldwide were married before 18, with nearly half of all child brides from South Asia [1]. Each year, approximately 15 million girls marry as children, which translates to one girl every two seconds [2]. While recent efforts have reduced child marriage rates, progress remains insufficient to meet the Sustainable Development Goal target of eliminating child marriage by 2030 [1]. Child marriage disrupts girls' lives, prematurely imposing adult responsibilities that can adversely impact their physical, emotional, and economic well-being. It interrupts education, limits earning potential, can lead to early motherhood, and restricts girls' agency, reinforcing cycles of poverty and gender inequality [3].

The Nepalese government is committed to ending child marriage, as evidenced by its signing of international treaties, such as the Convention on the Elimination of All Forms of Discrimination against Women, the Universal Declaration of Human Rights, and the Convention on the Rights of the Child. In Nepal, the legal age of marriage is set at 20 years for both men and women. Child marriage has been prohibited since 1963, with the 2015 Constitution reinforcing this prohibition [4]. However, enforcement remains limited, and child marriage persists as a long-standing social norm. It is estimated that Nepal has around 5 million women who were married as children, including 1.3 million who married before the age of 15 [5]. Nearly 35% of Nepalese women aged 20-24 were married by 18, compared to only 7% of men [6]. Despite the higher prevalence among women, the rate of child marriage among Nepalese men still ranks among the world's highest [7].

Several challenges hinder effective law enforcement in preventing child marriage at the local level in Nepal. These include persistent cultural beliefs, low awareness of legal consequences, social customs, and the practice of unregistered "love marriages" [8–10]. Additionally, the low rate of marriage registration complicates the identification and prevention of child marriages [11]. Law enforcement officials are often reluctant to prosecute impoverished families who may face further economic hardship from penalties, complicating legal deterrence efforts [12].

In Nepal, child marriage disproportionately affects women from marginalized ethnic groups, as well as those with lower levels of education, limited skills, and limited employment opportunities [12,13]. Parental education levels also play a key role, as child marriage is more common in families with lower educational attainment [14]. A recent evaluation of a program in Nepal highlighted addressing key structural obstacles, such as poverty and low education, as essential steps in reducing child marriage [15]. Financially, parents may view investment in a daughter's education as less valuable than in a son's due to patrilocal customs, where daughters move to their husband's family while sons remain to support aging parents [16].

Karnali, the largest and most remote province in Nepal, has some of the country's highest rates of child marriage. The provincial government has introduced initiatives to address this issue, such as the "Bank Account for Daughters: Lifelong Security" scheme, which incentivizes families to keep daughters in school and unmarried until age 20 by providing monthly savings that mature at adulthood [17]. Additionally, efforts are underway to educate religious figures, who often officiate weddings, on the importance of refusing to marry couples under the age of 20 [18]. However, data indicate that over half of women in Karnali marry before 18, and 18.8% of adolescent girls aged 15-19 have already given birth [19].

HealthBridge Foundation of Canada and International Nepal Fellowship are implementing a project titled Improving Reproductive Health and Preventing Child Marriage in Nepal and Vietnam. In Nepal, the project operates in Khadachakra municipality in Kalikot district. As

part of the project's baseline assessment, this qualitative study explores key community stakeholders' perceptions of child marriage causes and consequences to inform context-specific intervention strategies.

## Methods

### Study design and sampling

The study was conducted in Khadachakra municipality located in Kalikot district of Karnali province of Nepal. To understand the factors influencing child marriage, a range of stakeholders from different wards and government departments were identified and invited to participate in Focus Group Discussions (FGD) (n = 11) and Key Informant Interviews (KII) (n = 13).

The participants of the FGDs and KIIs were purposively selected from a range of community groups with a stake in the studied topic. Age, sex, location of residence, and profession were considered in selecting the groups and individuals to ensure a diversity of views. The FGD groups were identified by the local coordinator with the help of the research team, female community health volunteers (FCHVs), and schoolteachers. The inclusion criterion for the Key Informants (KI) was that they had worked in the municipality for at least six months. The inclusion criteria for the FGD participants were that they were residents of the Municipality and able to provide consent/assent. The local coordinator invited the FGD and KII participants in person or by phone, whichever was most convenient for the participants.

### Procedure

The study was a pragmatic study designed to address specific questions, and the design was in line with grounded theory to enable the exploration of social processes and interactions, and contextual factors that affect individuals' lives [20].

A topic guide was developed by the research team in Canada in collaboration with the local team in Nepal to reflect the study aim (S1 Text). The topic guide, information sheet, and consent form were translated to Nepali. NR, a Nepali national, and the local Project Officer, PH, reviewed and discussed the translated documents to ensure the original meanings were kept and culturally appropriate.

NR provided a two-day online training in Nepali for the research team consisting of two male and two female members. The research team facilitated two FGDs among their colleagues and a group of local residents in an adjacent community to test the procedure and the topic guide; the topic guide was finalized according to input from the FGD participants. The research team also received a refresher training from NR before they started data collection. The research team comprised members of the broader project implementation team and had formal training in health-related fields, bringing both technical knowledge and a deep familiarity with the community's social context. They adhered closely to a structured topic guide during data collection, ensuring that participants could freely express their views without any influence from leading questions. Additionally, regular debriefing sessions with the study team throughout data collection enabled continuous reflection, helping to address any emerging biases or challenges, and reinforcing the integrity of the data collected.

The FGDs and KIIs were conducted in June 2022. FGDs were conducted at schools and local gathering places, such as Health Mothers' Group meetings. KIIs were carried out at workplaces or other locations convenient for the interviewees. The FGDs and KIIs assessed views on child marriage status and trends, its perceived causes, and promising practices to reduce child marriage, and other Sexual Reproductive Health topics.

## Ethical considerations

The study protocol was approved by the Nepal Health Research Council (Ref. number 3092 dated 25 May 2022). Further, a letter of approval from the Khadachakra Municipality was obtained. The research team described the study objectives and data collection process to the participants, as well as to the parents/guardians of minors, using an information sheet. Participants and the parents/guardians of minors were offered an opportunity to ask questions to ensure they could make informed decisions regarding their own or their children's participation.

Formal written consent was obtained from all adult participants. Additionally, formal written consent was obtained from the parents/guardians of participants under 18 years old. The researcher verbally reviewed the consent form to ensure comprehension. Verbal assent was obtained from adolescent participants.

The KIIs and FGDs were conducted in a secure, private location to maintain confidentiality. Only the study team and participants were present during data collection, with no additional individuals involved. This setup ensured a private and comfortable environment for participants to share their experiences openly, safeguarding their confidentiality and supporting the integrity of the data collection process. Furthermore, prior to all FGDs and KIIs, explicit permission for audio recording was obtained from all participants.

## Inclusivity in global research

Additional information regarding the ethical, cultural, and scientific considerations specific to inclusivity in global research is included in the Supporting Information (S2 Text). We have adhered to the COREQ (Consolidated Criteria for Reporting Qualitative Research) guidelines to enhance transparency and rigor in reporting our qualitative research approach and results (S1 Table).

## Data analysis

All FGD and KII sessions were audio recorded and transcribed in Nepali. The transcripts were not returned to participants for comment or correction due to logistical constraints and confidentiality considerations. However, to maintain transparency and foster community engagement, summary findings were shared with the participants during a dissemination meeting. This allowed participants to review and discuss the general findings collectively, providing an opportunity for feedback and ensuring the study results were meaningfully communicated to the community. Transcripts were analyzed thematically. NR created a data sheet to organize the data by the questions in the topic guide including definition of child marriage, occurrence and pattern, consequences, perceived causes, and barriers towards accessing information and services. This step was purely for practical purposes to manage data in both Nepali and English. The data were grouped by FGDs and KIIs to identify any disagreement or agreement between different stakeholder groups and to specify the frequency of emerging themes. NR identified codes as they emerged using an iterative inductive process and recorded them in both the Nepali transcripts and the data sheet. BA then conducted further analysis using an iterative inductive process, working from the data sheet without the pre-identified codes from NR. During this process, NR assisted BA with any questions or clarifications needed to ensure an accurate understanding of the data nuances, particularly when meaning may have been lost in translation. After generating the codes independently, BA and NR reviewed the codes together, reflecting on any differences in the identified codes and considering how their understanding of the local context might influence data interpretation. While inter-coder reliability was not formally calculated, their discussions confirmed shared insights drawn from the data. This was followed by searching for themes, reviewing and refining themes, and

naming themes before writing up the findings. Additionally, PH, who was actively involved in data collection, reviewed the findings and confirmed agreement with the analysis.

The analysis framework was intended to follow Braun and Clarke's thematic analysis framework [21]. However, as noted earlier, the first step in their framework, familiarizing oneself with the data, was primarily undertaken by the Nepali investigator (NR), as they had access to the Nepali transcripts. In contrast, BA did not have direct access to the raw data (the Nepali transcripts) and worked with the prepared data sheet created by NR in English. Regarding the generating initial codes phase, NR generated codes from the Nepali transcripts, while BA used the data sheet to generate codes. This phase, as suggested by Braun and Clarke, was not fully adhered to; however, this approach did not result in discrepancies between the codes identified by the two authors.

Trustworthiness was ensured through a range of approaches. To establish credibility, the local team presented the findings to community stakeholders, including KII and FGD participants, who confirmed that the findings accurately represented their views. Dependability was maintained by systematically documenting each step of the data organization, translation, and analysis processes. BA and NR led the data management and met regularly to reflect on how their understanding of the local context might influence data interpretation. They worked closely with PH and another project coordinator to ensure confirmability. During the analysis the researchers actively looked for discrepancies between the views of different stakeholders; any divergence of views is reflected in the findings and discussion, however the data from all FGDs and KIIs were integrated and presented together.

## Results

### Description of FGDs and KIIs and participants

Eleven FGDs were held: two with Health Mothers' Groups, one with FCHVs, two with male community members, one with schoolteachers, one with schoolteachers and nurses, one with adolescent boy child club members, one with adolescent girl child club members, and two with Health Facilities Operations Management Committees with a total of 121 participants (51 adult men, 48 adult women, 12 adolescent boys, and 10 adolescent girls). Thirteen KIIs were undertaken with a range of stakeholders including local government employees, elected members of the municipality, local club leaders, and journalists (eight men, and five women). The mean age of adult and adolescent participants was 35.2 and 15.2 years, respectively. On average, the KIIs and FGDs lasted about 45 minutes and 69 minutes, respectively. The general characteristics of the FGD and KII participants are provided in the supplementary documents (S2 Table).

In this section, the findings from FGDs and KIIs are presented, covering the definition of child marriage, the significance of child marriage in the community, identified consequences of child marriage, and perceived causes of child marriage. A table detailing the themes and codes for consequences and causes of child marriage is provided in the supplementary documents (S3 Table).

### Definition of child marriage

All participants recognized a marriage at a young age as child marriage; while some groups characterized child marriage as any marriage under the legal marriage age of 20 years in Nepal, others defined it as marriages occurring under 18 years of age.

### The significance of child marriage in the community

Multiple participants in FGDs and KIIs agreed that child marriage has been declining in the community, particularly following the declaration to end child marriage in the municipality.

However, child marriage continues to be a substantial and often under-reported issue that requires attention. As explained by a male key informant "*We can say that the child marriage rate is very high in this area. [Khadachakra] municipality has done a lot of activities with adolescents related to preventing child marriage, still, adolescents are getting married before their 20s.*"

## Identified consequences of child marriage

Overall, both FGD and KII participants had a clear understanding of a range of adverse outcomes linked to child marriage. They identified the immediate and long-term consequences child marriage inflicts upon parents, particularly mothers, their children, and the broader community.

The main identified consequences of child marriage can be categorized into three overarching themes: maternal and child health, education and employment prospects for young parents, and marriage and gender-related issues. Almost all groups identified health consequences for mothers, encompassing concerns such as teenage pregnancies, unsafe abortion, uterine prolapse, physical debilitation, malnutrition including anemia, stillbirth, and compromised mental health. Similarly, they outlined adverse outcomes for children, including low birth weight, varied forms of malnutrition (underweight, wasting and stunting), and developmental disorders. The adolescent girls identified a lack of parenting skills and the low capacity of adolescent parents to nurture children as one of the main factors leading to the long-term consequences of child marriage. These consequences, they believed, adversely impact the growth and development of children over the long term.

A female KII participant highlighted the consequences of child marriage for mothers and children*: "First of all, they [young brides] are devoid of further education, and there are high chances of pregnancy immediately after marriage. Physically and mentally, they are not capable. So, they either perform unsafe abortions or they give birth to underweight newborns, posing the risk of death for both mother and their newborns.*"

All participants agreed on the substantial negative impact of child marriage on the couple's further education, employment prospects, and ultimately economic status, particularly for women. Teenage pregnancy emerged as a notable consequence of child marriage, disrupting further education of young mothers. Consequently, mothers with restricted access to educational opportunities and professional skills have reduced opportunities for employment.

Some FGD participants highlighted the long-term impact of child marriage on the fathers' employment opportunities. This is attributed to their role as the primary providers for their families, requiring them to assume familial responsibilities at a young age. A schoolteacher highlighted, "*Additional members [spouse and children] in the family require additional resources. Adolescents do not have a stable source of income, so they engage in daily wage jobs. Sometimes they sleep hungry as there is no money to feed themselves.*"

The FGD and KII participants also discussed marriage-related issues such as the high divorce rates among young couples, instances of polygamy, and experiences of Sexual and Gender-Based Violence. These issues were attributed to factors such as a lack of understanding of marital responsibilities, inadequate knowledge and skills to manage a family, and the absence of a stable income.

## Perceived causes of child marriage

Child marriage was found to be influenced by a complex interplay of factors operating at multiple levels. The factors identified in this study were organized and analyzed using the Social Ecological Model of Health Promotion (Fig 1), and the quotes are presented in Table 1. These levels encompass individual, interpersonal, community, and policy [22].

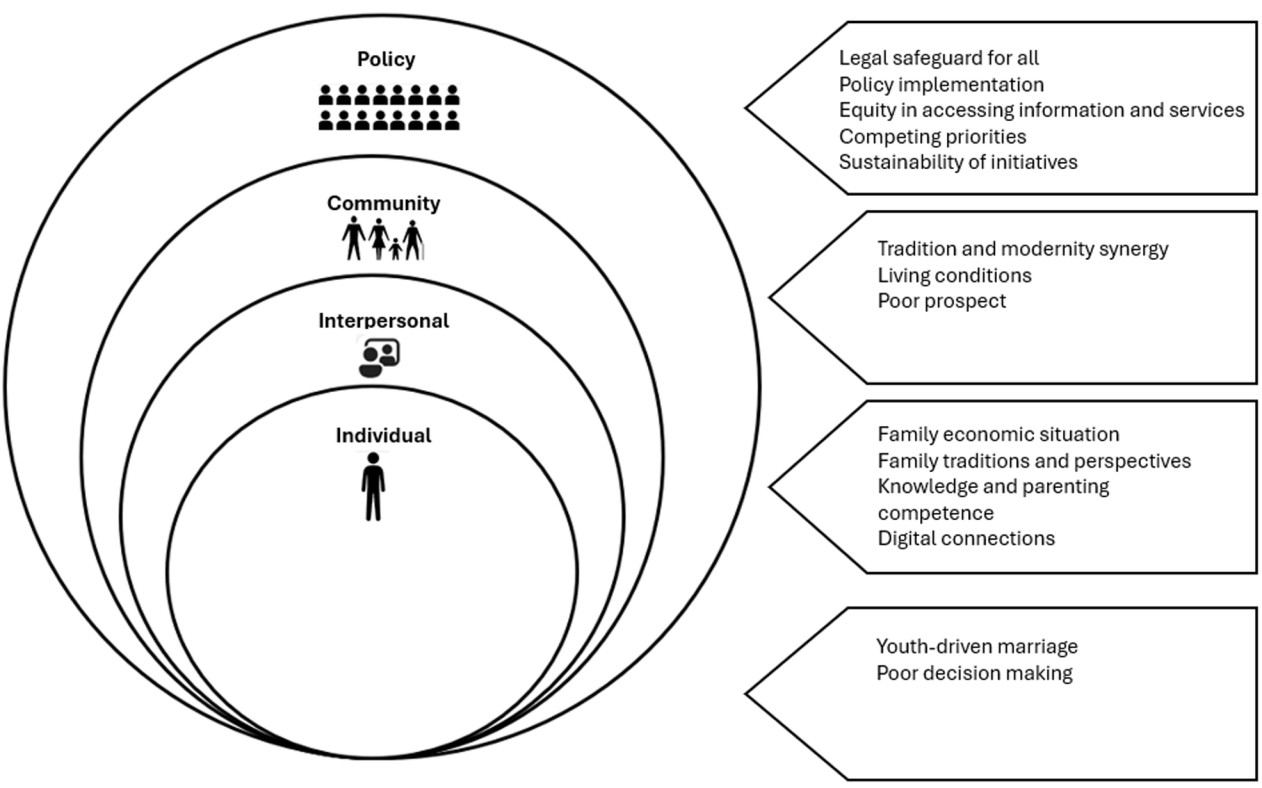

**Fig 1. Levels of perceived causes of child marriage by FGD and KII participants.**

## Individual level

Youth-driven marriages and adolescents' misperceptions about marriage were frequently highlighted as the primary catalyst for child marriage in recent years, as underscored by almost all adult KII and FGD participants.

There was discussion that marriage can be perceived as a means of escape when adolescent girls are exposed to domestic violence within their households, often stemming from issues such as alcoholism. Getting married for this reason fits under both family- related issues as well as personal issues.

## Interpersonal level

Different groups identified household poverty as a fundamental underlying factor in child marriage, manifesting in the form of large family sizes and the inability to afford education, particularly for girls. Parents sometimes advocate for their son's child marriage in order to get additional help with house and farm chores. Further, economic hardship motivates couples to have more children, assuming that once they reach adulthood, they will contribute to the family's income and provide financial support in their old age.

Intergenerational influences are also at play. Parents who, having themselves entered into marriage at young ages, perceive the child marriage of their offspring as customary and acceptable, and they encourage their children and grandchildren to marry early.

Multiple groups suggested that a lack of awareness about the consequences of child marriage among parents, coupled with inadequate parenting skills to provide guidance to

**Table 1. Perceived influence on child marriage-quotes from FGD and KII participants.**

| Contextual level | Themes and quotes |
| --- | --- |
| **Individual** | Youth-driven marriage |
| | "*These days, love marriages are very common, and the trend of arranged marriages is decreasing. Love marriages are happening at early ages [earlier than 20], and even the parents consent and accept these marriages.*" (KII_Male) |
| | "*Most adolescents get married by 15-16 years of age. Beyond this age, they consider themselves as too old to get married.*" (FGD_Schoolteachers) |
| | Poor decision-making |
| | "*Because of alcoholism, there is a quarrel in the family. Witnessing these, they [adolescent girls] feel sad and want to leave home to escape family tensions.*" (FGD_Schoolteachers) |
| **Interpersonal** | Family economic situation |
| | "*Girls especially get married earlier [than boys]. There are many problems in their house, and parents start treating them as a burden. Girls realize this and feel its time for them to leave home, so child marriages are happening in the community.*" (KII_Male) |
| | Family traditions and perspectives |
| | "*People do not feel child marriage is happening in their community. They treat all marriages as normal.*" (KII_Male) |
| | "*The elderly from households still desire to see the faces of their grandson soon [before they die] and thus insist on the early marriage of their son or daughter*". (KII_Female) |
| | Knowledge and parenting competence |
| | *Some [adolescent boys] get married due to family pressure. Parents insist their son get married early citing the need [of somebody] for household chores.*" (FGD_Health Mothers' Group) |
| | Digital connections |
| | "*Everything is available in cell phones: video calls, illicit materials etc. This is facilitating children to interact with each other, fall in love and marry early.*" (FGD_Schoolteachers and nurses) |
| **Community** | Tradition and modernity synergy |
| | "*In the past, parents used to arrange marriage for their adolescent children. Nowa-days, adolescents fall in love and initiate marriage by themselves.*" (FGD_FCHVs) |
| | "*They [community elders] say to get married when you are still young. So, there is a thinking to get married early.*" (FGD_Adolescent boy child club) |
| | "*The main problem lies within the community. When we talk to the community about minimizing child marriages, they [community members] agree and promise to support it. Later, when child marriage occurs in the community, they don't report it fearing negative relationships among community members. Thus, cases are never reported*". (KII_Male) |
| | Poor prospects |
| | "*There is nothing to do in the community. There are no educational opportunities, no play opportunities, no employment, and so because of poverty, they [adolescents] get married early.*" (KII_Male) |
| | "*Only legislations are not enough. Until adolescents have access to education and employment opportunities, only legal measures will not help.*" (KII_Male) |
| **Policy** | Legal safeguard for all |
| | "*NGOs and Community Based Organizations have done many programs related to child marriage [in our community]. We have declared our ward as a "child marriage-free ward." However, there still are incidences of child marriage in the community as these programs [related to preventing child marriages] have reached only a certain segment of the population. The poor and deprived people [in our community] are not yet reached by these programs*" (FGD_Health Mothers' Group) |
| | Policy implementation |

*(Continued)*

**Table 1.** (Continued)

| Contextual level | Themes and quotes |
| --- | --- |
|  | "*The government has allocated an annual budget for the capacity building of the girls, but the money is mostly being spent on celebrating cultural events. The budget should be directed towards other effective awareness-raising activities.*" (FGD_Health Facilities Management Committees) |
|  | Equity in accessing information and services |
|  | "*Adolescent girls are deprived of SRH information, the radio is mostly controlled by males in the households, girls and women listen to whatever is aired [they cannot change the channel]. In addition, women and girls are not given mobiles as there is a belief that they cannot handle the mobile properly and thus it would get destroyed easily.*" (KII_Male) |
|  | "*Mothers' Group is the major source of information for local women. However, not all eligible mothers are able to attend the Mothers' Group and so are deprived of the only source of health information* (FGD_Health Mothers' Group) |
|  | Sustainability |
|  | "*The government program on preventing child marriage is only one-time activities, there are no continuous activities to reinforce awareness*". (FGD_Health Facilities Operations Management Committees) |

adolescents, contributes to child marriage. In certain cases, parents have concerns that their daughters might elope independently, prompting them to initiate marriage into a reputable family instead.

The influence of mobile phones and social media in encouraging adolescents to engage in child marriage was a recurring topic of discussion among adult participants. In particular, easy access to mobile phones was recognized by adult participants as a facilitator for self-initiated child marriages.

## Community level

The findings indicate that the community is undergoing a cultural transition, characterized by the convergence of traditional viewpoints held by the older generation about marriage and the rising trend of self-initiated marriage among adolescents.

Mother's Group participants, among others, highlighted the influence of traditional healers and elders, and their strong opposition to change.

Primarily during the KIIs, participants identified living conditions and limited prospects as key catalysts for child marriage.

## Policy level

Across different community and stakeholder groups, Nepal's national policy against child marriage was identified as a crucial backbone to reduce child marriage. Local policies, such as the declaration of a child marriage-free community, were seen as key in translating national efforts into local action. However, many expressed the need for sustained, comprehensive, and inclusive programs, especially in remote and hard-to-reach areas where awareness and enforcement of policies are limited.

The perspectives of local authorities and community members sometimes diverged concerning implementing public policies. During KIIs, local officials provided examples of municipal efforts to combat child marriage, such as forming child networks within each ward to report on child marriages and establishing justice committees with representatives from law enforcement to handle complaints related to child marriage. Other initiatives highlighted included budget allocations for advocacy and awareness raising, alongside incentive programs,

such as setting up bank accounts for girls and contributing funds monthly on the condition that they remain unmarried until age 20.

The community members viewed these interventions as inadequate and ineffective in efforts to reduce child marriage. They provided specific instances where the local government overlooked opportunities to enforce the law by merely counselling the underage couple instead of pursuing legal action or subsequent monitoring.

Inequity in accessing information and programs was frequently highlighted as a significant barrier to reaching the zero-child marriage target in the community. It was discussed that programs are often not designed considering cultural appropriateness or the needs of women and girls, as well as their capacity to engage in these initiatives, especially if they are from lower socioeconomic backgrounds and residing in hard-to-access areas. Schoolteachers recognized the exclusion of boys from the majority of trainings and activities that predominantly target adolescent girls.

A consensus emerged that the efforts of local government and Non-Governmental Organizations (NGOs) have shown promising results. Inadequate resources and competing priorities were identified as impeding the local government's capacity to implement comprehensive and long-term interventions aimed at addressing child marriage.

Finally, the project's sustainability emerged as a recurrent theme. The participants discussed how NGOs established various youth clubs, child clubs, and gender-specific clubs in the past. The clubs operate effectively during the project's duration, but upon the project's conclusion, the clubs either collapse or their functionality declines. This situation deprives subsequent groups of reaping the project's benefits, and the initial cohort loses access to the project activities.

## Discussion

To our knowledge, this is the first qualitative study in Kalikot district of Karnali province in Nepal to explore the perceptions of different stakeholders about child marriage and its perceived causes and consequences. The FGD and KII participants are residents of Kalikot, and their view is valuable in understanding the issue of child marriage in this particular context, and for providing information that can be helpful in laying a foundation to develop effective and culturally appropriate interventions to reduce child marriage. The diversity of FGDs and KIIs provided a wide spectrum of viewpoints in understanding how the policies, initiatives and projects designed to reduce child marriage are perceived by different stakeholders.

Findings suggest that the participants were in agreement that child marriage is a gradually declining but still pressing challenge, and results in a number of negative consequences for young girls in particular. Because child marriage often leads to early child bearing, there are associated negative consequences for the young parents, especially the mothers, and their offspring, as well as for the community. Furthermore, participants expressed multiple views of the possible causes of child marriage that include individual, inter-personal, community, and policy level determinants.

Culture and gender norms, as well as the poor and isolated living conditions, were identified as interconnected and primary causes of child marriage. Inequity in accessing information and services was also discussed, especially for women and adolescent girls whose cultural needs may not be considered in the design and delivery of programs and interventions. The intersection of gender, socioeconomic status, and residential location can result in a significant proportion of adolescent girls and women not being reached by the services they need.

There are similarities between the identified causes of child marriage in this study and other studies from Nepal [8,9,23] and elsewhere [24–27]. Economic factors, social norms,

insufficient parental awareness, and constrained access to education are examples of similar emerging themes. However, the distinct theme that highlights the unique context of mountainous regions in Nepal is the geographic isolation of the communities, which limits access to economic and educational opportunities and leads to poor prospects, especially for those sub-groups of the community who live furthest from the community centre.

A common factor identified by adult participants was that widespread use of mobile phones, which facilitates easy connections among adolescents, along with easy access to online information. This exposes adolescents to content that can lead to poor decision-making, particularly in relation to their future, including marriage. Regarding social media, unlike adult participants, adolescents viewed open access to social media as a means of expanding their network and building strong relationships in their community. Despite the negative impact of digital technology and social media on adolescents' well-being [28,29], there is evidence of the positive impact of moderate and social use of social media for social connectivity and as a peer-to-peer support platform [30–32]. This needs further investigation to explore how the power of digital connection can be harnessed to increase awareness and knowledge of all community members, not only adolescents.

In Karnali, the rate of child marriage among adolescent girls surpasses that of adolescent boys (48% and 25%, respectively) [33]. Nevertheless, the high incidence of child marriage among adolescent boys indicates that a significant number of these marriages involve child couples rather than unions solely between an adult man and a child girl. The participants' perspective suggests diverse motivations that encourage marriages between adolescent boys and girls. Families of adolescent boys, driven by the prospect of additional assistance in the household, particularly in agrarian-dependent households, advocate for their son's early marriage. While the adverse effects of child marriage disproportionately impact girls, a few of the participants in this study highlighted the need to extend policies and protective measures to adolescent boys and to engage them in the interventions if reduction in child marriage is to be achieved.

The Nepal national government's strategy on ending child marriage [34] lays a strong foundation for eradicating child marriage, which is reflected in the declining pattern of child marriage in Nepal in recent years [6]. Our findings may suggest that in more traditional and isolated communities like Kalikot, the proximal influences such as family and community play a more influential role in comparison to macro-level influences (national policy and local government strategies). In these communities, a large portion of community members regard traditional healers as the primary health care providers, and religious leaders actively participate in overseeing arranged marriages by granting approval and determining suitable ceremony dates. Thus, it is imperative for any intervention addressing child marriage to incorporate their full involvement. Their engagement is critical in any interventions designed to reduce child marriage, and their role and responsibilities should be aligned with any attempts to achieve this.

Legal measures might yield minimal effectiveness in remote and isolated communities where tangible consequences to child marriage are lacking. The findings suggest that adolescents continue to marry through traditional ways, with families waiting until the young couple reaches the legal age before registering their marriage. Further, self-initiated marriages were identified as a growing trend in the community. In an environment where government-issued marriage certificates hold no relevance to daily life, the implementation of such policies has significant hurdles, indicating a requirement for the adoption of customized legal measures for these communities. The irrelevance of marriage certificates in these communities can explain the low number of registered marriages; a recent study by the Ministry of Health and Population in Nepal revealed that only 6.5% of married adolescents have registered their

marriage with civil authorities and obtained a marriage certificate. The same study suggests that marriage registration is lowest in rural Karnali province, which is our study site [6].

While ambitious programs such as the "Bank Account for Daughters: Lifelong Security" scheme, implemented by the Karnali provincial government [35], are crucial for addressing child marriage, it's essential to carefully evaluate their applicability in the local context. The scheme's eligibility criteria include registering within 45 days of a girl's birth, completing a high school education, and avoiding marriage until the age of 20. Considering the current rate of high child marriage, low levels of education, and limited low birth registration in the province [6] as well as social determinants such as peer pressure and poverty, it is improbable that the scheme's criteria will be met, making it unlikely to result in a reduction in child marriage rates; this aligns with observations from a similar conditional cash-transfer program to reduce child marriage in India [36].

Sustainability was identified as a key success factor for programs to work. The Nepal government facilitated the creation of a network of FCHVs; this network has contributed to providing reproductive age women with information and services on family planning, maternal, newborn and child health, and nutrition, among other topics [37,38]. Tackling child marriage requires an ongoing commitment through interventions that capitalize on government-supported initiatives. For instance, interventions that leverage available local resources, such as the FCHV-facilitated Mother's Group, are more likely to be sustainable [39].

One noteworthy finding of this study, which emerged through careful examination of the views of FGD participants, pertains to the perceived responsibilities and necessary actions defined by each group to prevent child marriage. While different groups recognized the important role of other stakeholders and discussed the actions that could be undertaken by these other groups, none of the groups acknowledged their own distinct roles. This phenomenon of "diffusion of responsibility" has the potential to result in a lack of initiative from any single group [40].

Although FGD participants acknowledged the local government's effort in dealing with child marriage, a significant proportion of adult FGD participants view the government as the principal stakeholder with the capacity to resolve the child marriage issue. The interplay of cultural norms, demography, and the type of governance could all contribute to shaping this view. As a result, the average community members may feel that their individual efforts would have little impact, as they perceive themselves as lacking resources and skills. These perceptions might also be attributed to the community's historical role in decision-making and initiating actions. One of the FGD groups highlighted a previous youth-led campaign that encountered considerable resistance from the community. Such past unsuccessful attempts to bring about change, known as "learned helplessness" in psychology [41], might also lead the community to not recognize their power to change the status quo.

Foremost, a gender transformative and long-term strategy should be pursued, one that considers cultural nuances and their implications in implementing successful legal measures. The Nepal government's strategy to end child marriage by 2030 [34] can serve as a basis for developing programs that are multifaceted, including infrastructure advancement, creating job opportunities, strengthening the healthcare system, and improving educational access for all sub-groups of the community. Failure to adopt such a holistic approach might lead to failure to overcome the challenge of child marriage, and could leave out two major escape routes for adolescents and young adults, particularly the more disadvantaged demographics: early marriage, which can worsen the situation, especially for girls with far-reaching inter-generational adverse impacts; or rural-out immigration, which could affect their future well-being and their home communities [42].

The study does have some limitations. Data collection was completed during the COVID-19 pandemic, and the principal investigators had to train the local data collection team, who lacked prior experience in qualitative data collection, through online platforms. As mentioned earlier, comprehensive training was delivered by the Nepali PI, NR, based in Ottawa. The data quality is considered satisfactory, and the feedback from facilitators suggests that because the research team was local and had cultural understanding, participants felt free to express their views. However, the absence of the experienced Nepali PI from the data collection process might have influenced the group dynamic, preventing probing questions and more in-depth discussions. Real-time access to data was not feasible, resulting in the PIs having to determine data saturation by consulting PH, the local Project Officer who facilitated five FGDs, interviewed 5 KIs, and attended two further KIIs as a note-taker. As the data analysis progressed, it became evident that data saturation had been achieved.

The FGDs and KIIs were conducted in Nepali. The non-Nepali PI, BA, leading the analysis and drafting the findings and discussions, could not access the verbatim transcriptions. Instead, a translated version of the organized data, prepared by NR, was used. Both PIs worked closely together, discussing the findings word by word to ensure the essence of the participants' perspectives was understood and analyzed accurately, thus overcoming the language barrier.

The extensive range of stakeholders in this study improved the data credibility; however, all FGD groups were homogeneous. Introducing a mix of stakeholders in an FGD could have sparked fresh and thought-provoking views. Further, the key role of traditional healers and religious leaders was frequently emphasized, but these figures were not among the FGD participants or KIs. However, the research findings were shared and discussed with local stakeholders, including some participants, to validate the findings and refine project activities. Specifically, tailored presentations were prepared by our local partners and delivered to traditional healers and religious leaders. These presentations emphasized the community's perception of them as influential figures in combating child marriage and encouraged them to take an active role as agents of change. The project team has been since collaborating with this group to explore various opportunities to leverage their influence.

Most KIs were male, reflecting the prevailing dominance of men in higher-level governmental positions. While this is not a study limitation, it could influence the perspective diversity.

This study provides an in-depth insight into child marriage in the context of remote mountainous areas in Nepal and lays a foundation for development of effective interventions in the future. A prominent emerging theme is the intersection of cultural norms and socioeconomic determinants, which contribute to the intricate nature of the child marriage issue. Another key theme is the significance of recognizing the agency of individuals, and community groups, in their ability to effect change. Understanding the roles and responsibilities of the community as a whole is crucial in order for the different players to take ownership of preventing child marriage and holding the local government accountable in their efforts.

Before concluding the discussion, it is crucial to outline tangible action items for donors, local government, and program managers. These recommendations capture the collective efforts needed to combat the complex issue of child marriage in Nepal and pave the way for sustainable change. Donors are urged to maintain financial support, emphasizing the declining trend in child marriage, with local stakeholders recognizing the role of interventions, among other factors, in this positive shift. While policymakers' efforts are acknowledged, the findings underscore the need for more robust legal measures to enforce child protection laws for all.

Program managers are urged to collaborate with funders, ensuring adequate time and resources, to observe a substantial decrease in child marriages through thoughtful planning in collaboration with the communities. Further, recognizing specific challenges faced by women, girls, and those in hard-to-reach areas to benefit from the available programs and services, the article highlights the necessity of engaging these groups to develop inclusive programs.

## Supporting information

**S1 Text. FGD and KII Topic Guide.**
(DOCX)

**S2 Text. Inclusivity in global research.**
(DOCX)

**S1 Table. COREQ Checklist.**
(DOCX)

**S2 Table. General characteristics of participants.**
(DOCX)

**S3 Table. Themes and codes.**
(DOCX)

## Acknowledgments

We sincerely thank Rebecca Brodmann for leading the conceptualization and funding acquisition of the project entitled "Improving Reproductive Health and Preventing Child Marriage in Nepal and Vietnam" which provided the context for the research. Special appreciation is also extended to Kamilla Pinter, the Project Officer, for her exceptional support throughout the implementation of this research. Our thanks also go to Moyandi Tamara Udugama for her contribution to data visualization. We express our appreciation to Dal Budha, the Project Coordinator at INF, for his unwavering support during the data collection phase and for his thorough review of the manuscript, and to Esther Gurung, for her ongoing support throughout the research. A profound acknowledgment is due to our dedicated local data collection team, Mahesh Sanjyal, Nabin Budha, and Sushmita Neupane, who demonstrated outstanding commitment and skill in navigating the mountainous and hard-to-reach areas and in using their knowledge of the culture and language to completing the assignment. Finally, we extend our gratitude to every individual who participated in this study. Their willingness to engage in discussions on this important and sensitive topic without any expectation reflects their commitment to improving their community. We deeply appreciate their contributions, recognizing that it is through collective efforts that meaningful progress is achieved. Artificial intelligence (AI), in the form of a language model, was used to improve the clarity of the language.

## Author contributions

**Conceptualization:** Nished Rijal, Behnoush Ahranjani, Sian Fitzgerald.

**Data curation:** Nished Rijal, Padma Hitan.

**Formal analysis:** Nished Rijal, Behnoush Ahranjani.

**Funding acquisition:** Sian Fitzgerald.

**Methodology:** Nished Rijal, Behnoush Ahranjani.

**Project administration:** Padma Hitan.

**Supervision:** Nished Rijal, Behnoush Ahranjani, Padma Hitan, Sian Fitzgerald.

**Validation:** Padma Hitan.

**Visualization:** Behnoush Ahranjani.

**Writing – original draft:** Behnoush Ahranjani.

**Writing – review & editing:** Nished Rijal, Padma Hitan, Sian Fitzgerald.

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
