## [Decision Letter · Decision Letter 0]

29 Sep 2024

PGPH-D-24-00451

Understanding the roots: local stakeholders’ insights on the causes and challenges in combating child marriage in mountainous Karnali, Nepal

Dear Dr. Ahranjani,

Thank you for submitting your manuscript to PLOS Global Public Health. After careful consideration, we feel that it has merit but does not fully meet PLOS Global Public Health’s publication criteria as it currently stands. Therefore, we invite you to submit a revised version of the manuscript that addresses the points raised during the review process.

Editor comments:

The reviewers and I found the manuscript to offer interesting and useful insights into the continued prevalence of child marriage in Nepal, drawing on focus group discussions with a wide range of community members and partners. However, aspects of the introduction, methods, and results require further revision to strengthen the manuscript.Specifically, please see Reviewer 1's suggestions concerning expanding the introduction. I would also encourage the authors to briefly situate child marriage in Nepal within the global literature of child marriage.Please see Reviewer 2's comments concerning the methods and results. In revising the methods and results, consulting a qualitative reporting guideline checklist (such as COREQ or SRQR) and ensuring alignment with standard reporting guidelines would be helpful. I agree with Reviewer 2 that providing the FGD guide(s) and final codebook as supplemental files would also be useful.

We look forward to receiving your revised manuscript.

Kind regards,

Marie A. Brault, PhD

Academic Editor

Journal Requirements:

2. Please provide a/amend your detailed Financial Disclosure statement. This is published with the article. It must therefore be completed in full sentences and contain the exact wording you wish to be published.

**Please only choose the relevant sentences from below**

1. Please clarify all sources of funding (financial or material support) for your study. List the grants (with grant number) or organizations (with url) that supported your study, including funding received from your institution. 

2. State the initials, alongside each funding source, of each author to receive each grant.

3. State what role the funders took in the study. If the funders had no role in your study, please state: “The funders had no role in study design, data collection and analysis, decision to publish, or preparation of the manuscript.”

4. If any authors received a salary from any of your funders, please state which authors and which funders.

3. In the online submission form, you indicated that "The data that support the findings of this study are available from the corresponding author upon request.". 

3. Uploaded as supplementary information.

4. Figure 1: Please confirm whether you drew the images / clip-art within the figure panels by hand. If you did not draw the images, please provide (a) a link to the source of the images or icons and their license / terms of use; or (b) written permission from the copyright holder to publish the images or icons under our CC-BY 4.0 license. Alternatively, you may replace the images with open source alternatives. See these open source resources you may use to replace images / clip-art:

- https://openclipart.org/

Additional Editor Comments (if provided):

Reviewers' comments:

Reviewer's Responses to Questions

**Comments to the Author**

1. Does this manuscript meet PLOS Global Public Health’s publication criteria ? Is the manuscript technically sound, and do the data support the conclusions? The manuscript must describe methodologically and ethically rigorous research with conclusions that are appropriately drawn based on the data presented.

Reviewer #1: Yes

Reviewer #2: Partly

2. Has the statistical analysis been performed appropriately and rigorously?

Reviewer #1: N/A

Reviewer #2: No

3. Have the authors made all data underlying the findings in their manuscript fully available (please refer to the Data Availability Statement at the start of the manuscript PDF file)?

Reviewer #1: Yes

Reviewer #2: No

4. Is the manuscript presented in an intelligible fashion and written in standard English?

Reviewer #1: No

Reviewer #2: Yes

5. Review Comments to the Author

Reviewer #1: Overall, this is a timely and necessary piece. My main comment is that the Introduction is quite brief and missing some key literature on child marriage prevention initiatives in Nepal, such as: Clark, C. J., Jashinsky, K., Renz, E., Bergenfeld, I., Durr, R. L., Cheong, Y. F., ... & Yount, K. M. (2023). Qualitative endline results of the tipping point project to prevent child, early and forced marriage in Nepal. Global Public Health, 18(1), 2287606. I think adding more information on what has been learned from previous attempts to reduce child marriage in the Introduction would also set up more suggestions for the Discussion. As a minor point, in the Methods it mentions "Additionally, formal writtenassent was obtained from the parents/guardians of participants under 18 years old." should be "formal written consent was obtained from parents/guardians and assent was obtained from children".

Reviewer #2: Context is very important for reporting qualitative results and the authors need to provide it e.g. life expectancy of birth is now highest in the Karnali region so the statement like "... the province lags behind national averages on various educational and development indicators ..." must be critically reviewed.

Line 59-60: Even though Nepal government has signed many international declarations to eliminate child marriage, its root cause at the local level (context) must be done competently. You must present them clearly in your findings.

Line 74-77: Since this study is done in one of the municipality of Kalikot district, its findings are not transferable, one of the key domain of trustworthiness, even to other districts of Karnali so the title must be changed accordingly.

Line 87-89: The deductive coding and its use in the analysis does not support the grounded theory.

Line 90-94: Provide the topic guide as supplement material of the manuscript for review

Line 95-101: Provide proof of credibility as the qualitative research must be an experienced one to conduct such trainings

Line 116-128: Explain why the data matrix code was created if this research is using a grounded theory approach. If some key issues are addressed in this paper then drop in from the description too so that it will not be considered as the possible salami slicing.

Line 116-128: Provide samples of transcripts of IDI and FGD with codes in as the supplement of the manuscript. Also describe how the validity of the transcripts were ensured.

Line 116-128: Since thematic analysis was done, describe which thematic analysis method (e.g. Braun and Clarke?) was used and what lens (gender?) was used as well.

Line 116-128: This section requires detail description of the rigor of qualitative data analysis based on the Trustworthiness (credibility, dependability, confirmability and transferability), of the process as well as findings. It must include description of reflexivity as well as bracketing, if any, used during the data collection and analysis phases.

Result section: Provide a table with description of IDI and FDG participants. Move the "Perceived causes of child marriage" theme in the beginning of the result section to provide the context. Explain why the socio ecological model was used only for this theme.

Result section: Explain why the institutional and policy level factors of the socio ecological factors are missing here. Provide key verbatim in the text as well. Table can be provided as an annex, do not report both in this section.

Result section: Provide a table with themes with the associated codes along with the measure of interceder reliability with justification.

Result section: Use "Consolidated criteria for reporting qualitative research (COREQ): a 32-item checklist for interviews and focus groups" to self-assess this manuscript. Describe it in the method section and also provide it as annex (supplement material)

Once these comments are addressed then review of discussion and conclusion will be meaningful.

6. PLOS authors have the option to publish the peer review history of their article (what does this mean? ). If published, this will include your full peer review and any attached files.

**Do you want your identity to be public for this peer review?** For information about this choice, including consent withdrawal, please see our Privacy Policy .

Reviewer #1: No

Reviewer #2: **Yes: ** Shital Bhandary

---

## [Decision Letter · Decision Letter 1]

17 Jan 2025

PGPH-D-24-00451R1

Understanding the roots: local stakeholders’ insights on the causes and challenges in combating child marriage in mountainous Karnali, Nepal

Dear Dr. Ahranjani,

Thank you for submitting your manuscript to PLOS Global Public Health. After careful consideration, we feel that it has merit but does not fully meet PLOS Global Public Health’s publication criteria as it currently stands. Therefore, we invite you to submit a revised version of the manuscript that addresses the points raised during the review process.

Editor comments:

The reviewers and I appreciate the authors' responsiveness to the previous reviews.Please note a few minor areas of clarification and edits that are needed, all in the methods.

We look forward to receiving your revised manuscript.

Kind regards,

Marie A. Brault, PhD

Academic Editor

Journal Requirements:

Additional Editor Comments (if provided):

Reviewers' comments:

Reviewer's Responses to Questions

**Comments to the Author**

1. If the authors have adequately addressed your comments raised in a previous round of review and you feel that this manuscript is now acceptable for publication, you may indicate that here to bypass the “Comments to the Author” section, enter your conflict of interest statement in the “Confidential to Editor” section, and submit your "Accept" recommendation.

Reviewer #1: All comments have been addressed

Reviewer #2: All comments have been addressed

2. Does this manuscript meet PLOS Global Public Health’s publication criteria ? Is the manuscript technically sound, and do the data support the conclusions? The manuscript must describe methodologically and ethically rigorous research with conclusions that are appropriately drawn based on the data presented.

Reviewer #1: Yes

Reviewer #2: Partly

3. Has the statistical analysis been performed appropriately and rigorously?

Reviewer #1: N/A

Reviewer #2: N/A

4. Have the authors made all data underlying the findings in their manuscript fully available (please refer to the Data Availability Statement at the start of the manuscript PDF file)?

Reviewer #1: (No Response)

Reviewer #2: No

5. Is the manuscript presented in an intelligible fashion and written in standard English?

Reviewer #1: Yes

Reviewer #2: Yes

6. Review Comments to the Author

Reviewer #1: No further comments except that "Verbal consent" should be changed to "Verbal assent" for adolescents.

Reviewer #2: You have reivewed well based on the previous comments. The absence of inter-coder reliability is still an issues here to address the dependability of this work. If a new inductive thematic analysis framework must be presented in the manuscript. It is still not clear how the trasncribed Nepali trascripts were translated and how its validity was ensured.

I would like you to declare the work done by all the authors so that the correct authorship can be awarded based on ICJME criteria.

7. PLOS authors have the option to publish the peer review history of their article (what does this mean? ). If published, this will include your full peer review and any attached files.

**Do you want your identity to be public for this peer review?** For information about this choice, including consent withdrawal, please see our Privacy Policy .

Reviewer #1: **Yes: ** Irina Bergenfeld

Reviewer #2: **Yes: ** Shital Bhandary

---

## [Editor Report · Decision Letter 2]

4 Feb 2025

Understanding the roots: local stakeholders’ insights on the causes and challenges in combating child marriage in mountainous Karnali, Nepal

PGPH-D-24-00451R2

Dear Dr Ahranjani,

We are pleased to inform you that your manuscript 'Understanding the roots: local stakeholders’ insights on the causes and challenges in combating child marriage in mountainous Karnali, Nepal' has been provisionally accepted for publication in PLOS Global Public Health.

Best regards,

Marie A. Brault, PhD

Academic Editor